# Predicting post-operative right ventricular failure using video-based deep learning

Rohan Shad [1,9], Nicolas Quach [1,9], Robyn Fong[1], Patpilai Kasinpila[1], Cayley Bowles[1], Miguel Castro[2], Ashrith Guha[2], Erik E. Suarez[3], Stefan Jovinge [4], Sangjin Lee[4], Theodore Boeve[4], Myriam Amsallem[5], Xiu Tang[5], Francois Haddad [5], Yasuhiro Shudo[1], Y. Joseph Woo [1], Jeffrey Teuteberg[5,6], John P. Cunningham [7], Curtis P. Langlotz [6,8] & William Hiesinger [1,6 ✉]

Despite progressive improvements over the decades, the rich temporally resolved data in an echocardiogram remain underutilized. Human assessments reduce the complex patterns of cardiac wall motion, to a small list of measurements of heart function. All modern echocardiography artificial intelligence (AI) systems are similarly limited by design – automating measurements of the same reductionist metrics rather than utilizing the embedded wealth of data. This underutilization is most evident where clinical decision making is guided by subjective assessments of disease acuity. Predicting the likelihood of developing post-operative right ventricular failure (RV failure) in the setting of mechanical circulatory support is one such example. Here we describe a video AI system trained to predict post-operative RV failure using the full spatiotemporal density of information in pre-operative echocardiography. We achieve an AUC of 0.729, and show that this ML system significantly outperforms a team of human experts at the same task on independent evaluation.

---

[1] Department of Cardiothoracic Surgery, Stanford University, Stanford, CA, USA. [2] Department of Cardiovascular Medicine, Houston Methodist DeBakey Heart Centre, Houston, TX, USA. [3] Department of Cardiothoracic Surgery, Houston Methodist DeBakey Heart Centre, Houston, TX, USA. [4] Department of Cardiovascular Surgery, Spectrum Health Grand Rapids, Grand Rapids, MI, USA. [5] Department of Cardiovascular Medicine, Stanford University, Stanford, CA, USA. [6] Stanford Artificial Intelligence in Medicine Centre, Stanford, CA, USA. [7] Department of Statistics, Columbia University, New York, NY, USA. [8] Department of Radiology and Biomedical Informatics, Stanford University, Stanford, CA, USA. [9] These authors contributed equally: Rohan Shad, Nicolas Quach. ✉email: willhies@stanford.edu

Predicting which patients will go on to develop RV failure after implantation of a left ventricular assist device has so far remained beyond the abilities of both human experts and existing automated algorithms[1–3]. A variety of clinical scoring systems have been developed with modest predictive power, with a maximum area under the receiver operating characteristic curve (AUC) of 0.60 in held-out datasets[1,4–7]. Without reliable methods to predict RV failure in the pre-operative setting, we have neither the means to decide in whom to aggressively intervene, nor an efficient method to randomize patients to trials that evaluate the efficacy of right ventricular treatment options. The gold standard method for determining which patients should receive advanced right ventricular support devices thus remains a 'clinical gestalt'[8], involving the patients' clinical course, lab parameters, and a qualitative assessment of myocardial function using a trans-thoracic echocardiogram. In this study, we describe an echocardiography machine learning system that enables time resolved characterization of motion parameters from each scan. We use this ML system to predict post-operative RV failure in LVAD (Left ventricular assist device) patients, using pre-operative echocardiograms alone. We compare the predictions of our ML system to those of contemporary RV failure risk scores, and further show that our ML system outperforms heart failure echocardiography experts in independent clinical evaluation. Figure 1 details an overview of the project.

In recent years, artificial intelligence has enabled automated systems to meet or exceed the performance of clinical experts across a range of image analysis tasks, from detection and diagnosis of disease to prediction of disease progression[9–13]. These systems typically draw conclusions from static images. Our video AI system processes two parallel spatiotemporal streams of data from echocardiography videos. The greyscale video channel and optical flow streams are combined within the convolutional neural network architecture with concatenation of activations prior to the terminal fully connected layers. We tested a variety of approaches, including various pre-training strategies, optimizers, input streams, and model architectures (Supplementary Table 7). Ultimately, we selected a three-dimensional 152-layer residual network for our echocardiography ML system as it gave the best performance. Architectural details and training strategy are detailed in the methods section. The area under curve (AUC) of the receiver operating characteristic (ROC) curve of our AI system, on the holdout testing dataset is shown in Fig. 2a. The ROC AUC for the AI system was 0.729 (95% CI 0.623–0.835; $n = 121$ patients; 327 scans). The corresponding Precision-Recall curves are shown in Supplementary Fig 4.

## Results

### Clinical decision-making workflow for end-stage heart failure.
Heart failure affects more than 6.5 million people in the United States alone, with an estimated 960,000 new cases diagnosed each year[14]. A heart transplant remains the gold standard for treating patients with end-stage heart failure. Demand, however, far outpaces the supply of transplantable hearts[15,16]. LVADs offer a mechanical alternative to transplantation, and the number of patients supported by these battery powered mechanical pumps have steadily grown[17]. In the contemporary era, an estimated 3500 LVAD implants are performed each year, approximately equal to the number of annual heart transplants in the United States[3,18]. Unfortunately, a third of all patients implanted with LVADs, develop a clinically significant degree of right ventricular failure (RV failure) soon after the procedure[4,19]. Underlying RV dysfunction and physiological changes under LVAD flow are both thought to contribute to the development of severe post-operative RV failure, which remains the single largest contributor to short-term mortality in this patient population[2,3,20,21].

An array of clinical biomarkers such as serum creatinine, alanine aminotransferase (ALT), blood urea nitrogen (BUN) are clinically used as surrogates of RV dysfunction[4,6,22]. Semi-quantitative

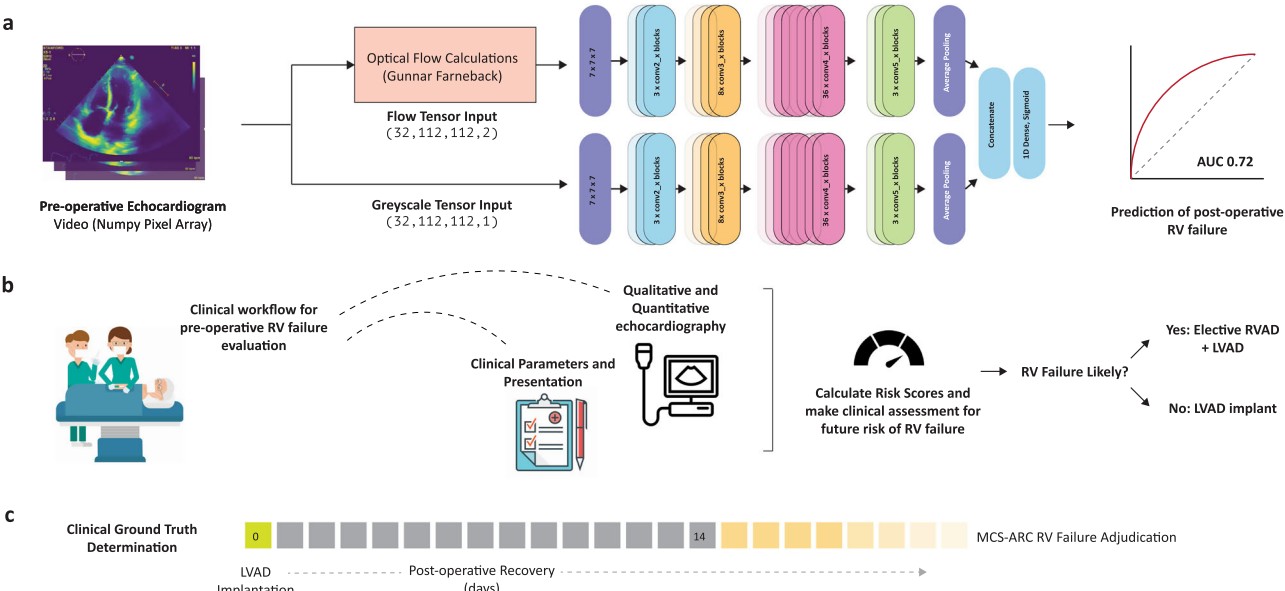

**Fig. 1 Overview of the project. a** Pre-operative echocardiography videos are processed as a stack of 32 frames. A two-stream implementation of raw greyscale videos and optical flow channels are fed into a 3D convolutional neural network to produce the prediction of RV failure. **b** The clinical workflow for predicting future risk of RV failure begins in the pre-operative phase using a combination of clinical parameters and a detailed echocardiographic assessment. Risk scores such as the CRITT and Penn scores are calculated thereafter to aid in risk stratification following which a decision is made to either electively implant a concomitant RVAD or proceed with LVAD alone. **c** The clinical ground truth is determined largely by the persistent need for inotropes past post-operative day 14 or right ventricular mechanical circulatory assist devices during the post-operative recovery period. MCS-ARC definitions are detailed in Supplementary Table 1. Artwork attribution from left to right in (**b**): Wikimedia Commons by Videoplasty.com; Anton Kalashny; Ralf Schmitzer".

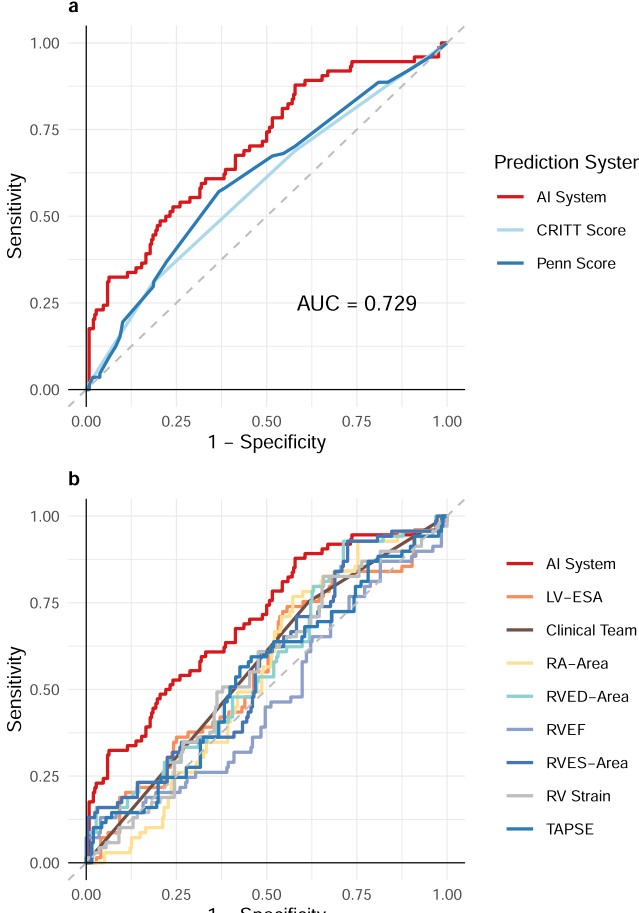

**Fig. 2 Performance of the AI system, clinical risk scores, and clinical benchmarking. a** ROC curve of the AI system compared to contemporary clinical risk scores. The performance of the AI system was 0.729 (95% CI 0.623–0.835). **b** ROC curves of clinical expert team and independently calculated metrics of right ventricular function compared to the AI system. The performance of the AI system was found to exceed both clinical experts and the traditional risk scoring systems. LV-ESA left ventricular end systolic area, RVED-Area right ventricular end diastolic area, RVEF RV Ejection Fraction, RVES-Area RV end systolic area.

echocardiographic measures and invasive hemodynamics are additional tools available to clinicians for the assessment of the right ventricle in the pre-operative setting. Echocardiographic measures such as TAPSE (tricuspid annular plane systolic excursion) and longitudinal strain are often used to characterize the severity of RV dysfunction. Patients with evidence of pre-operative RV dysfunction (via either echocardiography or invasive hemodynamic measurements) are typically hospitalized for aggressive pharmacological management and optimization[22]. Treatment regiments include diuretics to lower right ventricular preload, and inhaled drugs such as nitric oxide or phosphodiesterase inhibitors (PDE-5) to reduce right ventricular afterload[23]. Delaying LVAD implantation in a rapidly deteriorating patient is, however, not recommended, and often in such situations the risk of delaying surgery far outweighs the benefits of pre-operative cardiovascular optimization[24]. In the operating room, the LVAD implant proceeds via a median sternotomy, or thoracotomy with an additional smaller incision for aortic access. The LVAD is implanted into the left ventricular apex after coring out a segment of myocardium, and the synthetic aortic outflow graft is then sutured onto the ascending aorta. Intra-operative complications, although rare, may warrant reassessment of the right ventricle. Research has shown that

outcomes are more favorable following planned right ventricular assist device (RVAD) implantation, compared to emergent RVAD implantations where treatment is initiated after rapid deterioration of right ventricular function after initial LVAD implant[25]. The challenging prediction problem is thus compounded by the narrow therapeutic window available in the pre-operative phase.

**Multi-center clinical and echocardiographic RV failure dataset.** A dataset containing pre-operative and post-operative clinical variables along with paired trans-thoracic echocardiography studies was collected from three hospitals in the United States. The dataset consisted of 941 consecutive patients who had LVAD implants. 44 records were discarded due to missing data on duration of post-operative inotropes that prevented the adjudication of RV failure status. Data from an additional 173 patients were discarded because of missing apical 4-chamber echocardiograms or insufficient number of frames per video (Supplementary Fig. 3 and Supplementary Table 5). The dataset consisted of 159 (21.9%) females, and 562 (77.6%) male patients with an average age of 57.4 (sd 13.1) years. These figures are representative of the general LVAD patient population, and additional baseline characteristics and demographics by data split are outlined in Supplementary Table 3[26,27]. Most pre-operative scans were acquired at a median of 9 days (IQR 13 days) prior to LVAD implant. Patients undergoing surgery for a LVAD implant typically recover in the setting of a cardiac intensive care unit, where hemodynamic parameters (via invasive catheters) and clinical course determines the diagnosis of post-operative RV failure. We used the latest MCS-ARC (Mechanical Circulatory Support Academic Research Consortium) consensus definitions to grade each patient for post-operative RV failure to provide accurate and standardized clinical ground truth labels (Supplementary Table 1)[28]. Briefly, this incorporates invasive hemodynamic measurements, laboratory results for renal and hepatic function, and a persistent requirement for pharmacological inotropic support or right ventricular mechanical circulatory support devices. Training, validation, and holdout test datasets were randomly created to assess performance from the remaining 723 patients (1909 scans). Multiple scans were available from each patient, but no patients overlapped between the training, validation, and test datasets. One hundred eighty two patients (25.13%) were adjudicated to have post-operative RV failure.

**Echocardiography AI system and performance.** All current automated echocardiography systems—much like human echocardiography interpretations—are inherently reductionist in nature; a complex sequence and pattern of cardiac contraction is reduced to an outline of one or more chambers, from which a few global metrics of heart function are then calculated[29–31]. Quantifying subtle motion characteristics of the heart that predict future risk of disease requires a shift in approach to ML in echocardiography. We compared the predictive performance of our AI system against two popular clinical RV failure risk scores used for predicting post-operative RV failure—the CRITT score and Penn score[4,6]. These clinical risk scores combine clinical laboratory measures, hemodynamic readings, and qualitative assessments of cardiac function. The pre-operative variables used for calculating these scores are described in Supplementary Table 2[4,6]. The AUCs calculated for the Penn score (0.605; 95% CI 0.485–0.714) and the CRITT score (0.616; 95% CI 0.564–0.667) for our dataset are similar to previously published reports (Fig. 2a)[1,4]. We evaluated the ML system at both a liberal operating point of 80% sensitivity, where the specificity was 52.75%, and at a more conservative operating point of 80% specificity, where the sensitivity was 46.67%. Balancing the risks

and potential benefits of interventions in this patient population may require different thresholds of sensitivity and specificity. Observational data suggest that early and aggressive therapy with RVADs may improve survival in patients who are likely to develop post-operative RV failure[25]. These decisions must be made taking into account the significant additional morbidity associated with biventricular mechanical circulatory assist[21,26,32]. Right ventricular assist devices potentially could thus be offered based on cutoffs from the conservative operating point of our ML system. An increasing proportion of patients implanted with LVADs today are listed as 'destination therapy' candidates, where the expectation is that the LVAD will provide lifelong circulatory support[26]. A heart transplant on the other hand may be offered after the initial LVAD implant as a 'bridge to transplant' strategy, with priority for receiving an organ guided by prevailing UNOS (United Network for Organ Sharing) allocation policies. Anecdotally 50% of all surviving LVAD patients at 5-years are bridged to a heart transplant in our current dataset[33]. Although not all LVAD patients are suitable candidates for a heart transplant, a more liberal threshold for the ML system may be used to inform listing strategy for heart transplantation owing to the poor short and long-term survival of LVAD patients who go on to develop post-operative RV failure.

**Clinical benchmark with human experts**. We benchmarked the performance of our AI system against a clinical heart failure echocardiography team. In clinical practice, echocardiographic assessment is used as an important adjunct metric to gauge the likelihood of downstream RV failure. Numerous metrics such as TAPSE, right ventricular diameters, fractional area change, and right ventricular longitudinal strain have all been described as potential predictors for post-LVAD RV failure[34–38]. While decisions are rarely based on echocardiography alone, the combination of poor clinical presentation and 'severely depressed' preoperative RV function may be predictive of post-operative disease[22]. A blinded clinical benchmark has to our knowledge, not been conducted for this clinical problem. The team of board-certified cardiologists and a sonographer with 12 years of clinical sonography experience were blinded patient outcomes, and graded scans independently from the remainder of the research team. A web-based annotation tool (MD.ai, New York, NY) was configured for measuring echocardiographic metrics of right ventricular function[38]. In addition to quantitative metrics, the clinical team also identified the patients they predicted would go on to develop RV failure after the operation. These measures were calculated for all 121 patients in the testing dataset ($n = 91$ controls; $n = 30$ cases with RV failure) and an additional subset of 86 ($n = 70$ control; $n = 16$ cases) randomly selected patients from the validation set. The AUCs of the manually extracted metrics ranged between 0.525–0.571. The AI system ($n = 121$; test set results only) outperformed both clinical readers and all quantitative echocardiographic metrics. The best performing manual echo metric (RV longitudinal strain) had an AUC of 0.5623 (95% CI 0.464–0.660; ΔAUC for AI system 0.167 (95% CI 0.159–0.175, $p = 0.025$) (Fig. 2b). The AUC of the clinical team predictions was AUC of 0.579 (95% CI 0.4971–0.643); ΔAUC for AI system 0.159 (95% CI 0.126–0.192), $p = 0.016$). The clinical reader team had a specificity of 37.89% and a sensitivity of 76.09%; for the same sensitivity, the specificity of the AI system was 54.95%.

**Saliency maps and visualizations**. Interpretability of clinical AI systems has implications in identifying failure modes as well as in establishing trust and confidence in end users[39,40]. In this paper we utilized gradient backpropagation to generate saliency maps[41]. These are computed based on the imputed gradient of the target output with respect to input, where only non-negative gradients are backpropagated; in non-technical terms, the goal of this technique is to find input data that would exemplify the features the network uses to predict RV Failure (or lack thereof). We show that for each patient, regions of activation were localized exclusively to the myocardium and valves. The cardiac chambers (ventricles and atria) themselves showed no activation. Furthermore, motion characteristics of specific regions of the heart contribute towards the prediction of RV failure at different phases of the cardiac cycle (Fig. 3). In patients where the AI system correctly predicted RV failure, saliency maps localized over the right atrium and the region of the tricuspid annulus. Similarly, the AI system correctly predicted the absence of RV failure when activation was localized over the right ventricle and right atrium. In cases where the AI system appeared to rely heavily on the interatrial and interventricular septum, the quality of predictions seemed to decline. While septal motion aberrations have often been described as a predictor of RV failure in the clinical setting, it may also be a feature of acute isolated RV volume overload—a challenging overlap in presentation on echocardiography[42].

## Discussion

In this study we demonstrate a machine learning system capable of characterizing subtle myocardial motion aberrations on echocardiography for downstream clinical analyses. We utilize this system to predict the outcome of post-operative RV failure in heart failure patients under consideration for LVAD implant, and show that our ML echocardiography system outperforms board-certified clinicians equipped with both manually extracted echocardiographic metrics and state-of-the-art clinical risk scores. Our algorithm predicts a binary outcome of RV failure, though our methods can readily be extended to predict continuous and multi-class outcomes of interest.

The poor predictive performance of contemporary clinical risk scores is well documented. Many of the input variables are consequences rather than true predictors of RV failure. Most risk scores were developed without internal cross-validation or falter when evaluated on held-out datasets[1,5,8,43]. More recently, some investigators have attempted to use Bayesian networks on pre-operative parameters sourced from the INTERMACS registry to predict post-operative RV failure[44]. Critically, pervasive issues with missing data and severe class imbalance in these registries (2.7% RV failure, vs 97.3% normal patients) may have biased the results with overestimations of predictive power, especially when using performance metrics sensitive to changes in class imbalance[45]. We have employed the latest standardized definitions of post-operative RV failure[28]. In the past, definitions of post-operative RV failure were largely based on the need to implant a right ventricular assist device—an intervention driven by surgeon preference and institution specific nuance. The current definitions allow for standardized and generalizable ground truth labels for post-operative RV failure across all participating institutions. By abstaining from defining 'mild' and 'moderate' RV failure (as only more 'severe' grades were found to impact long-term outcomes), the current MCS-ARC guidelines offer a more clinically relevant target for our ML system.

All contemporary echocardiography ML systems rely on supervised segmentation algorithms to outline cardiac chambers[29,46,47]. Most recently, Ouyang et al. described a weakly supervised video segmentation system to calculate ejection fraction using left ventricular tracings in conjunction with spatiotemporal convolutions[30,47]. Our methods offer a number of key improvements: First, instead of segmenting cardiac chambers, our AI system directly analyses spatiotemporal information from the cardiac musculature and valves by default—the principal regions of interest

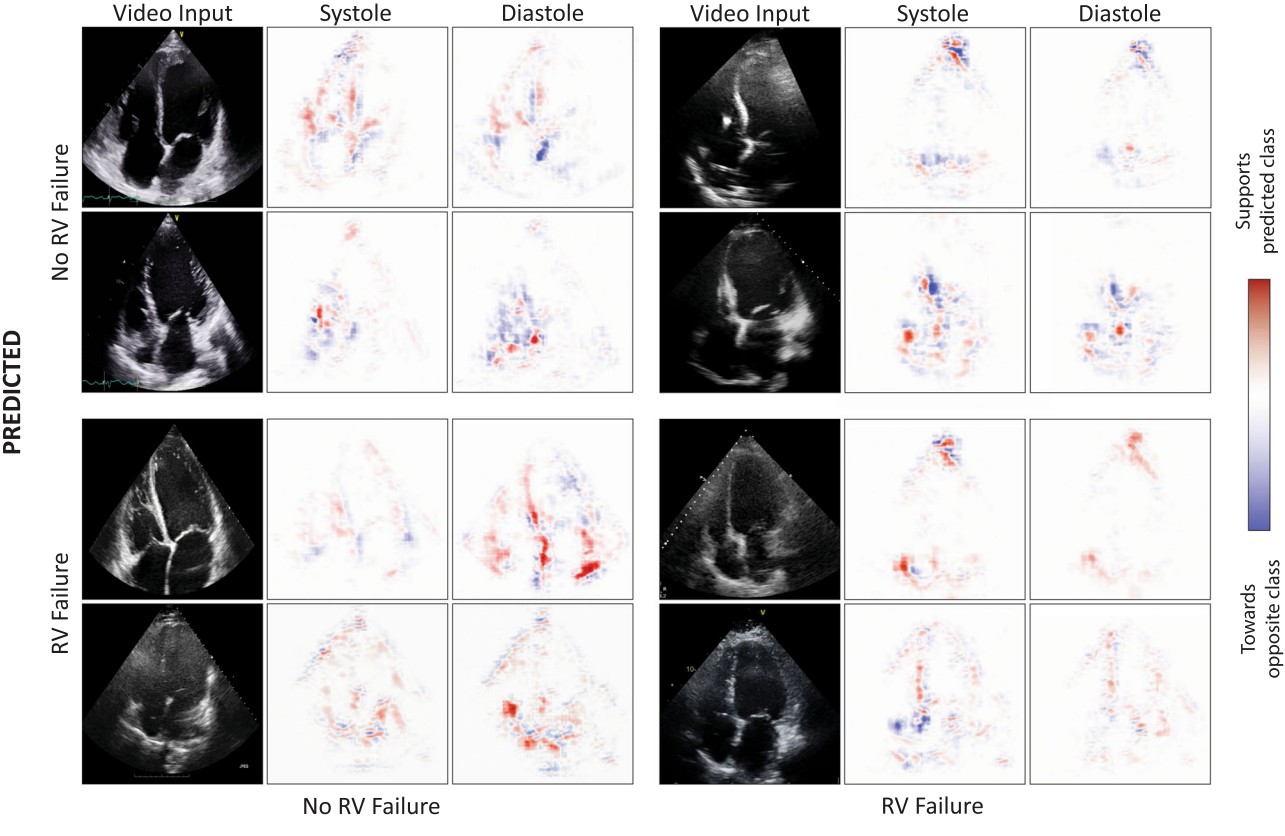

**Fig. 3 Analysis of saliency maps from pre-operative echocardiograms.** Representative input videos and visualizations for both systolic and diastolic phases of the cardiac cycle across patients with and without RV failure, in the form of a confusion matrix. True positives (bottom right quadrant), false positives (bottom left quadrant), true negatives (top left quadrant), and false negative examples (top right quadrant). Colour scale for each quadrant represents regions that contributed most to the predicted class (red) and those that pushed predicted probability away from the predicted class (blue).

in all cardiac diseases. This enables the algorithm to characterize subtle, regional aberrations in myocardial motion, that traditional manually extracted echocardiographic measures fail to capture. Secondly, our end-to-end system tracks features of importance without human supervision in the form of segmentation masks. This method is not dependent on cardiac view plane or chamber, enabling rapid deployment of our methods to a diverse array of echocardiography problems. Finally, we use two streams of spatiotemporal information in the form of greyscale video channels and optical flow to directly predict downstream outcomes of interest. Combined two-stream networks achieve state-of-the-art performance on large video recognition datasets[48]. Our system makes inferences on a single study within 500 ms on a single Nvidia GeForce RTX 2080Ti GPU. An additional computational overhead of 10 s, however, is needed for calculating optical flow per input video. In the context of our clinical problem, the additional computational overhead of calculating optical flow is acceptable. Future work may focus on integrating faster deep learning methods of optical flow estimation within the AI pipeline for applications that require real time inference[49].

Analyzing echocardiography videos rather than clinical or laboratory metrics allows for a direct visual assessment of cardiac function. The literature surrounding the predictive value of manually calculated metrics of cardiac function, however, remains inconclusive[37,50]. Our echocardiography ML system outperforms manually calculated metrics of myocardial function in predicting RV failure. Automating the calculations of these hand-measured metrics using image segmentation algorithms are therefore unlikely to be predictive of our outcome of interest,

further supporting our rationale for transitioning to an end-to-end architecture. While our AI system was trained using the largest echocardiographic heart failure dataset of its kind, there remain several limitations of our work. The retrospective nature of our dataset and the lack of a standardized echocardiography acquisition protocol limits the quality and timing of the scans prior to the index operation. Most scans were taken 9 days prior to LVAD surgery, though this window was larger for some patients. Furthermore, a comprehensive assessment of right ventricular function remains challenging without additional echocardiographic views (subcostal views, parasternal long axis)[51]. Madani et al. describe an image-based neural network for automated echocardiography view classification that could be integrated into a multi-view RV failure risk prediction system, and it is plausible that superior performance can be attained using prospectively collected echocardiographic data from multiple view planes[52]. Key to such efforts are standardized and comprehensive protocols for acquisition of scans at pre-defined timepoints before index surgery. Prospective evaluation in the clinical setting will be essential to understand the limitations of our technology. We found that pre-training with large video datasets (Kinetics-600) was critical for model performance[53]. The same could not be said for pre-training from the Echo-Net Dynamic dataset despite sharing the same imaging modality. While beyond the scope of this current manuscript, future attempts at self-supervised pre-training with cardiovascular imaging datasets may yield superior results[54,55]. Finally, unlike hard radiological or histopathological ground truth labels, the clinical ground truth for post-operative RV failure leaves room

for subjectivity despite the MCS-ARC guidelines. As part of additional experiments, we evaluated the incorporation of certain baseline demographic and clinical features (Gender, INTER-MACS score, Creatinine, ALT) into the training loop. We did not see a meaningful improvement over our best performing video-based system, although the improvements were more pronounced in some of the shallower neural networks evaluated (Supplementary Table 7). Additional hemodynamic information in the form of invasive pressure telemetry may further improve the performance of multi-modal AI systems integrating health record information and imaging.

Our methods are, to our knowledge, the first to predict the onset of disease using video-based echocardiography ML. Our system may serve as a clinical decision support system beyond instituting effective RV rescue treatments for this patient population; including the early detection of left heart failure, disease phenotyping, and a multitude of cardiac clinical decision support applications where treatment or patient selection is guided by qualitative echocardiography assessments.

## Methods

**Data sources and study population**. Data in the form of clinical outcomes and raw echocardiography DICOM files were sourced from the departments of Cardiothoracic Surgery at Stanford University (CA), Spectrum Health (MI), and the Houston Methodist Hospital (TX); (IRB 52440) with a waiver of consent owing to the retrospective nature of the research. All patients aged 18 years or older with at least one pre-operative trans-thoracic echocardiogram as well as a complete pre-operative and post-operative assessment of RV failure during index-hospitalization as per the MCS-ARC consensus definitions (Mechanical circulatory assist academic research consortium) were included (Supplementary Table 1)[28]. Apical 4-chamber views for trans-thoracic echocardiograms taken closest to the day of surgery were used for this study. Our final dataset comprised 723 patients. Raw data was anonymized and linked to clinical outcomes data via a unique study-ID. As the MCS-ARC definitions were standardized in August 2020, we manually reviewed each patient record for the duration of inpatient admission to collect pre-operative and post-operative clinical parameters following a pre-determined and standardized protocol[28]. This enabled accurate grading of RV failure severity along with the calculations of the various RV failure risk scores. The research team tasked with developing and training the artificial intelligence system did not have access to the original patient charts. Clinical data were stored and managed in REDCap[56].

**Outcomes**. The primary outcome of the study was the ability of the AI system to identify and predict the likelihood of post-operative RV failure using only pre-operative trans-thoracic echocardiograms as the input. All patients with MCS-ARC defined post-operative RV failure were included in the 'RV failure' group ($n = 182$ patients; 25.13%); the remainder were kept as controls ($n = 541$). Multiple scans were available for each patient (501 RV failure (26.24%); 1408 controls).

**Data pre-processing**. Echocardiograms were first de-identified by stripping all private health information (PHI) from file metadata and by obscuring any sensitive information in the videos. The complete removal of all sensitive information was verified manually on all videos before proceeding to downstream postprocessing. Areas outside of the scanning sector were masked to remove any miscellaneous markings in the video frames that may otherwise influence the neural networks. The videos were then normalized by dividing each pixel value by the pixel of maximal intensity. The frames of the processed videos were additionally downsampled by bi-linear interpolation to a $112 \times 112$ resolution, and all videos were temporally normalized to framerate prior to training and evaluation. Additional experiments with an input resolution of $224 \times 224$ pixels were also conducted on shallower residual networks. Optical flow was calculated prior to model training using an OpenCV implementation of the Gunnar Farnebäck method based on polynomial expansion[57]. Additional data augmentation operations were performed on each frame of the videos in the following order as part of the training loop: random rotation up to 10°, random brightness multiplications, and random 2D shearing.

**Neural network architecture and training**. We use a three-dimensional convolutional neural network[58], built using the Keras Framework with a TensorFlow 2.1 (Google; Mountain View, CA, USA) backend and Python, that tracks motion features and structural features in blocks of 32 consecutive frames. We make use of bottlenecked residual blocks expanded to three-dimensions. We used validation data to do model selection over a range of architectures and tested multiple 3D neural network architectures before selecting a two-stream fusion 152-layer 3D Residual Network with bottlenecks incorporated within the residual blocks (Supplementary Fig. 2)[59,60]. The residual blocks utilize a convolutional layer with a

$3 \times 3 \times 3$ kernel, sandwiched between two $1 \times 1 \times 1$ convolutional layers. The first convolutional layer utilizes a $7 \times 7 \times 7$ kernel[60]. The network weights were initialized using the Xavier normal initializing scheme, and was optimized using the AdamW algorithm[61,62]. We incorporated clinical and demographic variables into the vision network by concatenating a 1-D vector to the terminal fully connected layer of each residual network. We showed marginal improvements over several pure vision-based neural networks in Supplementary Table 7. The network was trained for 50 epochs on a batch size of 8, with an initial learning rate of $1 \times 10^{-5}$. Training was stopped early if the training loss did not improve for 5 epochs. For each echocardiogram, 5 random 32-frame clips of the full movie were subsampled and passed through the trained neural network. The average of the 5 outputs was calculated to predict RV failure. Hyperparameter tuning was carried out on the validation dataset. We implemented a proportional loss weighting strategy during training with a binary cross-entropy loss function, to account for the effect of minor class imbalance. Major python packages used in this work include numpy (1.18.1), opencv-python (4.5.1), scikit-learn (0.22.2), and pROC package (1.17) and dplyr (1.0.7) in R.

**Pre-training**. All candidate networks were pre-trained on the Kinetics-600 dataset for video action recognition[53]. Videos in the Kinetics-600 dataset were converted to greyscale and optical flow and subsampled for 32 consecutive frames prior to pre-training. Pre-training on Kinetics-600 was performed on servers, each with eight NVIDIA V100 GPUs, on the Stanford Sherlock Supercomputing Cluster. We additionally experimented with transfer learning from networks initialized on the EchoNet Dynamic dataset for Ejection Fraction prediction with over 10,000 Apical 4-Chamber echocardiography videos[30]. The results were, however, similar to those of randomly initialized networks. Training was stopped when validation loss did not improve, and the model weights were saved. The networks for RV failure prediction were finally initialized with these weights and the terminal linear activation function was replaced by a sigmoid function. In our experiments we find that pre-training significantly improves training convergence, with higher validation AUC and lower cross-entropy losses. This corroborates findings from a number of groups working on both medical imaging and general purpose video action recognition problems[11,30,48].

**Visualizations and interpretations**. We used an implementation of Gradient backpropagation to generate saliency maps for the AI system as it makes predictions of RV failure outcome when passed through the three-dimensional convolutional neural network[39,41].

Visualization of representation learned by higher layers of the network are generated by propagating the output activation back through the ReLU function in each layer $l$ and setting the negative gradients to zero:

$$R_i^{(l)} = (f_i^l > 0) \cdot \left( R_i^{l+1} > 0 \right) \cdot R_i^{l+1} \quad (1)$$

Alternative visualization techniques such as layer wise relevance propagation were also considered. The neurons that contribute the most to the higher layers receive the most 'relevance' from it. The relative contribution of each pixel towards the final predicted value is quantified to satisfy the following equation:

$$R_{i \leftarrow k}^{(l,l+1)} = R_k^{(l+1)} \frac{a_i w_{ik}}{\sum_h a_h w_{hk}} \quad (2)$$

The total relevance $R$ is conserved between layers $l$. During each forward pass, neuron $i$ inputs $a_i w_{ik}$ to the next connected neuron $k$. The messages $R_{i \leftarrow k}^{(l,l+1)}$ distribute the relevance $R_k^{(l+1)}$ of a neuron $k$, onto the preceding neurons that feed into it at layer $l$. The presence of skip connections in 3D residual networks violates the assumptions of relevance conservation, limiting us to Gradient backpropagation.

**AI performance and comparison with clinical risk scores**. The US dataset was split in an approximate 66:17:17 ratio into a training, validation, and test dataset. The stratified split ensured proportional numbers of unique patients with and without RV failure in each group. The validation set was used for hyperparameter tuning and an ensemble of three models trained with identical settings were used to generate final predictions at the scan level. On freezing the model weights, model performance was evaluated on the testing dataset using the area under curve (AUC) of the receiver-operator characteristic and AUC of the precision-recall curve. We further compared the predictive performance of our AI system against clinicians equipped with two contemporary risk scores used in for predicting post-operative RV failure—the CRITT score and Penn score. The variables used for calculating these scores are described in Supplementary Table 2[4,6]. Missing data prevented the calculation of clinical risk scores for as many as 37% of all patients in our total dataset. For this reason, we elected to use a multiple imputation strategy following Rubin's rules to pool and calculate AUCs[63,64]. No significant difference in AUC was noted in the original incomplete dataset vs the imputed dataset (1% lower in imputed Penn Score; 0.2% higher in imputed CRITT Score). Imputation diagnostic plots are shown in Supplementary Fig 5.

**Comparisons with manually calculated echocardiography metrics**. We compared the performance of AI based RV failure prediction to a set of manually

derived echocardiographic measures of right ventricular function. These measures were independently calculated by two board-certified cardiologists for 207 patients ($n = 161$ controls; $n = 46$ with RV failure) in our dataset, using metrics of RV function previously described[38]. The echocardiography scans were hosted on a secure Google Cloud Bucket at full resolution, ranging from a resolution of ($422 \times 636$) to ($768 \times 1024$) pixels. A custom cloud-based annotation system (MD.ai, New York) was used to allow the clinical heart failure team to remotely take measurements, perform quality control, and adjudicate metrics of RV failure (Supplementary Fig. 1). The scans were uploaded in a random order and no time constraints were provided to the clinical team. The clinical team-based adjudication approach mimics the clinical setting, where clinical sonographers and board-certified cardiologists together assess patients with end-stage heart failure. A full list of echocardiographic measurements is detailed in Supplementary Table 6 and Supplementary Fig. 6.

**Statistical analyses**. No statistical methods were used to predetermine sample size. To evaluate the stand-alone performance of the AI system, ROC curves were calculated as empirical curves in the sensitivity and specificity space. AUCs for ROC curves were computed with trapezoids using the pROC package[65]. AUC for the Precision-Recall curve was computed using the interpolation method described by Davis and Goadrich[66]. To compare the performance of our AI system against clinical risk scores and manually calculated echo metrics, we calculated non-parametric confidence intervals on the AUC using DeLong's method[67], following which $p$ values were computed for the mean difference between AUC curves. Missing data for clinical risk scores were addressed with multiple imputation with chained equations using a univariate imputation method (predictive mean matching)[68]. The pooled AUC for the imputed datasets ($n = 20$; max iterations = 50) was calculated using Rubin's Rules by log transforming the AUC prior to pooling[63]. Statistical analyses were conducted in R (v3.6.2).

**Reporting summary**. Further information on research design is available in the Nature Research Reporting Summary linked to this article.

## Data availability

The dataset for this study was acquired under data transfer agreements that restrict public release due to the potential embedded protected health information present in the raw data. Access to subsets of the data can be obtained as follows: Please contact A.G. and S.L. for access to paired raw imaging and patient level data from Houston Methodist and Spectrum Health, respectively. Please contact A.G. (gashrith@houstonmethodist.org) and S.L. (Sangjin.Lee@spectrumhealth.org) for access to paired raw imaging and patient level data from Houston Methodist and Spectrum Health, respectively. Access may be granted for research use subject to institutional ethical approvals. Please contact W.H. (willhies@stanford.edu) for access to raw imaging data and patient level data from the Stanford cohort. Access to all Stanford subsets (training/validation/testing) will be granted to all accredited researchers pending approvals of relevant data use agreements via Stanford's office of sponsored research. The EchoNet Dynamic dataset is publicly available at: https://echonet.github.io/dynamic/. The Kinetics action recognition dataset is publicly available at: https://deepmind.com/research/open-source/kinetics.

## Code availability

TensorFlow codebase including our models, training and evaluation scripts, and accessory R scripts for the final analyses and plots are available on GitHub at: https://github.com/rohanshad/postop_rv_failure_echo.git[69].

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

## Acknowledgements

Some of the computing for this project was performed on the Stanford Sherlock and Nero clusters. We would like to thank Stanford University and the Stanford Research Computing Center for providing computational resources and support. This project was in part supported by a Stanford Artificial Intelligence in Medicine and Imaging (AIMI) Seed Grant, and Cloud Compute Credits from Google Cloud. R.S. was supported in part by the American Heart Association Postdoctoral Fellowship Award (Grant #834986).

## Author contributions

R.S. and W.H. designed the experiments, and wrote the manuscript along with C.L. The codebase was authored by R.S. and N.Q. Computational experiments were performed by R.S. and N.Q. under the supervision of C.L., J.P.C., and W.H. Clinical datasets were curated by R.F., R.S., and J.T. P.K. and C.B. assisted with quality control for RV failure grading in the multi-center datasets. A.G., M.C., E.E.S., S.L., T.B., and S.J., provided external datasets and expertise on final manuscript. T.X., M.A., and F.H. assisted with the independent clinical benchmarking study. J.T., Y.S., and Y.J.W. provided sections on clinical impact in the final manuscript. The work was supervised by W.H.

## Competing interests

The authors declare no competing interests
