## [Peer Review File · Nature Communications]

Reviewers' Comments:

Reviewer #1:

Remarks to the Author:

I read with interest the manuscript entitled "Predicting post-operative right ventricular failure using video-based deep learning" by Shad et al. submitted for publication in *Nature Communications*.

The authors developed a novel video-based AI system from pre-operative echocardiography scans of patients treated with left ventricular assist device implantation. The algorithm is trained to predict post-operative right-sided heart failure, using the full spatiotemporal density of information from echo scans. The authors reported an AUC of 0.729, specificity of 52% at 80% sensitivity and 46% sensitivity at 80% specificity. Furthermore, the authors reported that the ML system significantly outperforms a team of human experts tasked with predicting right-sided heart failure on independent clinical evaluation. The authors suggest that, the methods are generalizable to any cardiac clinical decision support application where treatment or patient selection is guided by qualitative echocardiography assessments.

This is an important work; indeed, the prediction of postoperative right-sided heart failure is difficult and thereby, most of clinical prediction scores are modest at best. I have the following comments:

1. First: nonetheless, preoperative factors possibly play a major role in the development of a manifest right-sided heart failure following LVAD surgery, the rationale of predicting post-LVAD right-sided heart failure solely based on preoperative data is inadequate. Several perioperative factors (particularly those happening during and after surgery), including cardiotomy, LVAD settings, interventricular interdependence forces, outflow graft position, and postop complications are known factors to contribute to occurrence of post-LVAD right-sided heart failure.
2. There are potential other preoperative markers that could possibly strengthen the predictive capacity of the proposed AI video algorithm. Commonly used lab markers could improve the prediction algorithm.
3. Additional value of physiology information: would the addition of flow feature to the proposed algorithm improved the predictive power for right-sided heart failure?
4. Saliency maps and the echo views: typically, a right ventricle focused view is required for accurate RV functional and volumetric assessments per guidelines. The authors could elaborate on the performance difference of the tested algorithm using standard versus focused RV view.
5. It is important to clearly describe the clinical expert tools for the assessment of RV function/size as a competing index for prediction of RV Failure. Clinical experts replies on integrated approach including size, function of the RV including global and regional strain, tricuspid annular systolic excursion ...etc.
6. There is another preprint paper by the same authors (doi: <https://doi.org/10.1101/2020.05.05.20092494>), where the authors reported AUC for the AI system trained using the Stanford LVAD dataset of 0.860 using pre-operative echocardiograms alone. The authors need to elaborate on these different algorithms. The ensemble model of the 3D CNN and the unsupervised improved dense trajectory system with a supervised classifiers seems superior to current one in this manuscript regarding prediction of postop RV failure.

Reviewer #2:

Remarks to the Author:

The paper presents a deep learning system for predicting post-operative right ventricular failure (RV failure) using pre-operative echocardiography scans. The video-based deep model is trained end-to-end to directly predict the RV failure based on the spatiotemporal information of the echo videos.

The main strengths of the paper are:

- The manuscript is generally well written and properly organized.
- A very interesting clinical problem is tackled.
- The system outperforms a team of human experts in the task of predicting RV failure based on echocardiograms.
- The presented details of the system in the manuscript is adequate and the methodology has high reproducibility.

However, the major limitations and possible spaces for improvement of the paper are:

- The methodology is not fairly quite technically novel. The video-based deep models are extensively explored in the literature. In specific, two-stream video processing using optical flow as a secondary network input is previously explored in computer vision (e.g. Fig 2-e in [1]) and also in echo imaging [2]. Investigation of more recent trends in deep learning models for video analysis, such as [3, 4, 5] could add technical value to the paper. Analysis of recent deep learning Spatio-temporal models is given in [6].
- The paper aims to predict RV failure solely based on pre-operative echo videos (apical 4-chambers view). In this setup, while the method outperforms clinical experts, it seems still the obtained accuracy may not be high; sensitivity is less than 50% at 80% specificity (reported AUC of 0.729, specificity of 52% at 80% sensitivity and 46% sensitivity at 80% specificity). It can be understood that the accuracy limitation could be due to the highly challenging nature of the tackled problem. The authors may further clarify the importance and clinical applicability of the obtained results.
- There is the possibility that more information may be needed to properly predict the RV failure. Possible improvements in the results may be obtained by investigating the incorporation of more information into model input, such as the addition of other echo views and other relevant info from the patient's health record.

Other minor comments:

- The input to the network is 32 frames of the video. The authors could clarify if the input videos are trimmed or re-sampled to 32 frames? If they are trimmed, what is the used criterion for selecting the input temporal window?
- Adding a color legend to Fig. 3 could improve the clarity of the figure.
- The "LVAD" acronym is first used in line 45 and later defined on line 68.

Best regards,

Mohammad H. Jafari

Ph.D. Candidate,

Electrical and Computer Engineering Department

University of British Columbia (UBC), Vancouver, Canada

References:

[1] Carreira, Joao, and Andrew Zisserman. "Quo vadis, action recognition? a new model and the kinetics dataset." proceedings of the IEEE Conference on Computer Vision and Pattern Recognition. 2017.

[2] A Unified Framework Integrating Recurrent Fully-Convolutional Networks and Optical Flow for Segmentation of the Left Ventricle in Echocardiography Data,
https://link.springer.com/chapter/10.1007/978-3-030-00889-5_4

[3] SlowFast Networks for Video Recognition,
https://openaccess.thecvf.com/content_ICCV_2019/papers/Feichtenhofer_SlowFast_Networks_for

_Video_Recognition_ICCV_2019_paper.pdf

[4] Video Action Transformer Network,
https://openaccess.thecvf.com/content_CVPR_2019/papers/Girdhar_Video_Action_Transformer_Network_CVPR_2019_paper.pdf

[5] X3D: Expanding Architectures for Efficient Video Recognition,
https://openaccess.thecvf.com/content_CVPR_2020/papers/Feichtenhofer_X3D_Expanding_Architectures_for_Efficient_Video_Recognition_CVPR_2020_paper.pdf

[6] Deep Analysis of CNN-based Spatio-temporal Representations for Action Recognition,
<https://arxiv.org/pdf/2010.11757.pdf>

Reviewer #3:

Remarks to the Author:

The authors present a proper study of echocardiogram videos for an interesting prediction task: post-operative RV-failure from pre-operative data. The results while modest in terms of AUC, greatly outperform baselines presented. The methods seem to be an interesting contrast to Ouyang et al. Overall, I would recommend the paper for revision before acceptance.

Comments:

- It would be nice to expand on Fig1b to explain to the audience how manual systems are currently employed in the clinical decision making workflow. As this is Nature Comms and not a Cardiology-specific journal, we shouldn't assume such knowledge. A natural question is how could the manual systems perform so poorly as standard-practice and what expectations should the reader have in terms of reasonable latent patterns in the data that one would expect could be useful for prediction?
- The authors mention "We tested a variety of approaches, including various pretraining strategies, 56 optimizers, input streams, and model architectures." But there's no discussion on what was actually attempted and comparative results. In particular, the input streams + model architectures are of most interest.
- The authors are directly feeding multiple frames as a 0-th dimension of the tensor to the ResNet. While Ouyang has a similar approach with highly curated inputs. The standard practice within video-processing in deep learning and computer vision has shown the importance of including some modeling inductive bias (e.g. recurrent neural network variants or positional encoding in transformers) to leverage the concept of time.
- What type of data curation and data augmentation is or is not performed? this could greatly improve results and affect adoptability implications
- Captions need to be revised. They're currently very broadly written and not explanatory enough. authors should include key active takeaways and point out / explain the most relevant comparisons/acronyms/names
- There needs to be more analysis broadly speaking. In particular Figure 3 could benefit from additional analysis.. for example there could be a confusion matrix to peer into when/why the model and manual systems are right vs wrong (with some interpretation as well)

minor comments:

- Figure 1 caption needs to be revised to have a one-liner followed by A and B.
- Improve background to cite related work in deep learning and echo:
- Madani, Ali, et al. "Fast and accurate view classification of echocardiograms using deep learning." NPJ digital medicine 1.1 (2018): 1-8.
- Ghorbani, Amirata, et al. "Deep learning interpretation of echocardiograms." NPJ digital medicine 3.1 (2020): 1-10.
- Wonder if it makes more sense to pretrain on EchoNet dynamic dataset instead to enable better transfer learning. for discussion on usage of natural images for medical imaging prediction tasks, refer to Ke, Alexander, et al. "CheXtransfer: performance and parameter efficiency of ImageNet models for chest X-Ray interpretation." arXiv preprint arXiv:2101.06871 (2021).

Reviewer #4:

Remarks to the Author:

I have experience using optical flow/two stream networks in echocardiography also, and actually my personal experience was the optical flow aren't didn't add that much beyond either 3D CNNs or recurrent convolutional CNNs. Did the authors investigate other forms? My suspicion is that the optical flow algorithms start to break down a bit due to significant echocardiographic noise and the general textured nature of the myocardium.

In the data augmentation, the authors say they used "3-dimensional shearing". Do they mean they used 2d shearing on each frame, or they literally sheared in 3 dimensions? The latter sounds incredibly disruptive, as you'd be moving spatial data from 1 frame into another, essentially introducing wild levels of dyskinesia, with systole in 1 part of the heart moving into a diastolic period of another, for example. "Normal" hearts are unlikely to look normal after that, surely, and picking up the subtle signs of septal motion which they postulate are important but be next to impossible?

I'm surprised they had to use videos as small as $112 * 112$ pixels, as the original Quo Vadis paper many years ago I think used similar sized videos, and obviously GPU sizes have progressed significantly since then. I suppose this is because the authors used a VERY deep (152 layers) ResNet rather than the VGG networks/I3D used previously. I personally wonder if using a less deep network but with higher resolution might be better, and would be interested to hear if the authors tried this.

My personal feeling is standard formulae such as BCEloss do not need to be given in a manuscript where 1) anyone who is interested in that level of depth would already know it, 2) it's not the focus of the paper. But I don't feel too strongly.

My personal feeling is pretraining on non-medical datasets before transferring to completely different modalities is overhyped and in my experience helps little, versus just good initialisation, See <https://arxiv.org/abs/1902.07208>. But I accept this is a somewhat controversial view and appreciate most reviewers will expect it. I am therefore interested to hear the authors found it useful.

Reviewer #1 (Remarks to the Author):

I read with interest the manuscript entitled “Predicting post-operative right ventricular failure using video-based deep learning” by Shad et al. submitted for publication in Nature Communications.

The authors developed a novel video-based AI system from pre-operative echocardiography scans of patients treated with left ventricular assist device implantation. The algorithm is trained to predict post-operative right-sided heart failure, using the full spatiotemporal density of information from echo scans. The authors reported an AUC of 0.729, specificity of 52% at 80% sensitivity and 46% sensitivity at 80% specificity. Furthermore, the authors reported that the ML system significantly outperforms a team of human experts tasked with predicting right-sided heart failure on independent clinical evaluation. The authors suggest that, the methods are generalizable to any cardiac clinical decision support application where treatment or patient selection is guided by qualitative echocardiography assessments.

This is an important work; indeed, the prediction of postoperative right-sided heart failure is difficult and thereby, most of clinical prediction scores are modest at best. I have the following comments:

1. First: nonetheless, preoperative factors possibly play a major role in the development of a manifest right-sided heart failure following LVAD surgery, the rationale of predicting post-LVAD right-sided heart failure solely based on preoperative data is inadequate. Several perioperative factors (particularly those happening during and after surgery), including cardiotomy, LVAD settings, interventricular interdependence forces, outflow graft position, and postop complications are known factors to contribute to occurrence of post-LVAD right-sided heart failure.

This is an important and clinically very relevant comment. The aim of this project was primarily to see if severe post-LVAD right heart failure could be predicted before a patient is taken into the operating room. Reliable estimates drawn prior to taking the patient into the operating room could help dictate the institution of invasive right ventricular mechanical assist at the time of index LVAD operation – whether ECMO or percutaneous systems.¹ Planned institution of mechanical RV support has been shown to result in superior outcomes, possibly by addressing RV failure within a narrow early therapeutic window. This window may be missed when relying on biochemical surrogates of the post-operative phase. Finally, post-operative complications were not included in the models simply because the most impactful clinical decision window for risk stratification via our algorithms exists in the **pre-operative** setting.

A key distinction must be drawn between a ‘predictor’ vs a ‘consequence’ of RV failure. Findings such as worsening renal function after LVAD implantation may be downstream consequences of RV failure that has already begun to manifest as end-organ dysfunction, rather than a true ‘predictor’. As you very correctly pointed out in your comment, complications may arise later in the post-operative phase (when the therapeutic window may have already passed). As others have elaborated in recent editorials (for prediction of other surgical problems), there is scarce clinical utility in predicting post-operative RV failure at this juncture using such variables.² Often patients will have some degree of mild right ventricular dysfunction that is amenable to intensive medical management, though there is no reliable way to quantify interventricular dependence forces in clinical practice outside of qualitative echocardiography assessments. Outflow graft positioning theoretically can cause extrinsic compression of the right ventricle, though this **exceedingly** rare in clinical practice. The outflow graft itself is more likely to be occluded / kinked by such mal-positioning given the compliant Dacron material it is fabricated from. Finally, all LVAD patients receive a cardiotomy to the apex of the left ventricle. We don’t expect that our system can account for all potential complications that may occur in the course of treatment. Instead we hope to screen for patients who are qualitatively sicker at baseline in terms of right ventricular status for potential intervention.

This comment does, however, raise interesting possibilities for future research. It is possible that we may be able to model the ‘additive risk’ of a patient via investigations, imaging, and telemetry taken at different time intervals along the course of hospitalization. Such approaches have been used for the development of early warning

systems for renal failure using multilayered encoders and recurrent neural networks for tabular data from electronic health records.³ This remains beyond the scope of this current manuscript, as we aimed to create a pre-operative risk stratification system rather than a continuously active prediction system.

2. There are potential other preoperative markers that could possibly strengthen the predictive capacity of the proposed AI video algorithm. Commonly used lab markers could improve the prediction algorithm.

Thank you for this insightful comment. To incorporate clinical biomarker and demographic information into the AI system, we incorporated the following: Gender {int}, admission INTERMACS score {int}, and pre-operative serum creatinine {float} and ALT {float}. The integer {int} variables were used as is, the numeric {float} variables were scaled and normalized across the entire dataset. We then concatenated a vector of dimensions {batch size, 4} containing this information to the terminal fully connected layer of 3 residual network architectures. The networks were trained as described elsewhere in the manuscript. We describe the results on the validation set below, and in Supplementary Table 7. The improvements (validation set) were most obvious in the smaller residual networks with benefits that tapered off with deeper ones.

Model	AUC (95% CI)
Resnet-18 + EHR	0.644 (0.525 – 0.762)
Resnet-50 + EHR	0.682 (0.617 – 0.746)
Resnet-152 + EHR	0.739 (0.642 – 0.837)

3. Additional value of physiology information: would the addition of flow feature to the proposed algorithm improved the predictive power for right-sided heart failure?

The echocardiography data collected as part of routine clinical care did not always contain doppler flow measures. Point measurements of hemodynamics available from right heart catheterization have not shown any meaningful predictive value alone or in combination with clinical parameters in risk scoring systems (Figure 2a of our paper). However, it is plausible that multi-modality ML – incorporating raw pressure waveforms from invasive right heart catheterization and EKG waveforms with echocardiography for example – may improve the ability of neural networks to better capture embeddings representative of this patient population. Such telemetry data was also not available for this retrospective dataset. Continuing from the above comment we have however amended the discussion in Paragraph 4 to read as follows, as the question of hemodynamics is a very pertinent one:

“As part of additional experiments, we evaluated the incorporation of certain baseline demographic and clinical features (Gender, INTERMACS score, Creatinine, ALT) into the training loop. We did not see a meaningful improvement over our best performing video-based system, though the improvements were more pronounced in some of the shallower neural networks evaluated (Supplementary Table 7). Additional hemodynamic information in the form of invasive pressure telemetry may further improve the performance of multi-modal AI systems integrating health record information and imaging.”

4. Saliency maps and the echo views: typically, a right ventricle focused view is required for accurate RV functional and volumetric assessments per guidelines. The authors could elaborate on the performance difference of the tested algorithm using standard versus focused RV view.

Thank you for this insightful comment. One of the goals of our study was to ensure the data we used was relatively standardized across the entire cohort. In standard clinical practice LVAD patients have historically been assessed with a qualitative ‘severity’ grading for their right heart function using a limited transthoracic echocardiogram. While the British guidelines encourage the use of RV focused views for accurate RV assessments, the ASE guidelines list the RV focused views as alternatives to the standard apical 4 chamber view.^{4,5} However, all guidelines currently recommend using multiple views to improve the assessment of the right ventricle, as each view may offer complimentary information.⁶ We have amended the discussion section (Paragraph 4) addressing

some of these issues:

“Furthermore, a comprehensive assessment of right ventricular function remains challenging without additional echocardiographic views (subcostal views, parasternal long axis).⁵¹ Madani et. al describe an image-based neural network for automated echocardiography view classification that could be integrated into a multi-view RV failure risk prediction system, and it is plausible that superior performance can be attained using prospectively collected echocardiographic data from multiple view planes.⁵² Key to such efforts are standardized and comprehensive protocols for acquisition of scans at pre-defined timepoints before index surgery.”

We agree that additional subcostal views and parasternal views may have offered additional information that may have been useful towards predicting downstream RV failure. In clinical practice, measurements of TAPSE and semi-quantitative assessments of tricuspid valve function are routinely performed. Some of these variables are used in the clinical risk scoring systems, and we additionally show the predictive performance of TAPSE in Figure 2b (above). Quantifying the true difference in performance using standard views vs RV focused views is not feasible with our study design. The number of training examples would change if we were to re-train our algorithms using just standard or RV focused views, and this would directly impact performance (given the size of our dataset that we are working with). We feel it is important to note here that despite these limitations, this is the largest echocardiographic dataset for LVAD patients in the world that we know of.

5. It is important to clearly describe the clinical expert tools for the assessment of RV function/size as a competing index for prediction of RV Failure. Clinical experts replies on integrated approach including size, function of the RV including global and regional strain, tricuspid annular systolic excursion ...etc.

We have added a separate subsection detailing the clinical workflow and current gold standard for how patients are evaluated for downstream risk of RV failure. In this section we describe the steps used by cardiologists and cardiac surgeons in assessing each patient at the time of admission for LVAD implant and the factors that

determine the decision to implant mechanical circulatory systems for additional right ventricular support. This subsection is titled: “**Clinical decision-making workflow for end stage heart failure**”, and now reads:

“An array of clinical biomarkers such as serum creatinine, alanine aminotransferase (ALT), blood urea nitrogen (BUN) are clinically used as surrogates of RV dysfunction.^{4,6,22} Semi-quantitative echocardiographic measures and invasive hemodynamics are additional tools available to clinicians for the assessment of the right ventricle in the pre-operative setting. Echocardiographic measures such as TAPSE (tricuspid annular plane systolic excursion) and longitudinal strain are often used to characterize the severity of RV dysfunction. Patients with evidence of pre-operative RV dysfunction (via either echocardiography or invasive hemodynamic measurements) are typically hospitalized for aggressive pharmacological management and optimization.²² Treatment regimens include diuretics to lower right ventricular preload, and inhaled drugs such as nitric oxide or phosphodiesterase inhibitors (PDE-5) to reduce right ventricular afterload.²³ Delaying LVAD implantation in a rapidly deteriorating patient is however not recommended, and often in such situations the risk of delaying surgery far outweighs the benefits of pre-operative cardiovascular optimization.²⁴ In the operating room, the LVAD implant proceeds via a median sternotomy, or thoracotomy with an additional smaller incision for aortic access. The LVAD is implanted into the left ventricular apex after coring out a segment of myocardium, and the synthetic aortic outflow graft is then sutured onto the ascending aorta. Intra-operative complications, though rare, may warrant reassessment of the right ventricle. Research has shown that outcomes are more favorable following planned right ventricular assist device (RVAD) implantation, compared to emergent RVAD implantations where treatment is initiated after rapid deterioration of right ventricular function after initial LVAD implant.²⁵ The challenging prediction problem is thus compounded by the narrow therapeutic window available in the pre-operative phase.”

We additionally define how these echocardiographic metrics are calculated in the caption for Supplementary Table 6. The caption now reads:

“Area measurements were made in the appropriate phase of the cardiac cycle via manual segmentation in the apical 4-chamber view. TAPSE It is measured as the length between the end-diastolic and peak systolic points of the lateral tricuspid annulus. RV longitudinal strain was calculated from mid-endocardial end-diastolic and end-systolic manually traced lengths of the RV free wall, and expressed as absolute values, as previously described.”

6. There is another preprint paper by the same authors (doi: <https://doi.org/10.1101/2020.05.05.20092494>), where the authors reported AUC for the AI system trained using the Stanford LVAD dataset of 0.860 using pre-operative echocardiograms alone. The authors need to elaborate on these different algorithms. The ensemble model of the 3D CNN and the unsupervised improved dense trajectory system with a supervised classifier seems superior to current one in this manuscript regarding prediction of postop RV failure.

The preprint linked is a very early version of our work using just our internal Stanford dataset for a conference submission. In that preprint, we did not have a separate hold out dataset (only internal cross-validation) and some of the specific methods described (dense trajectories) ultimately did scale well when we were finally able to secure a larger more diverse multi-center dataset. We are not planning to submit that work as a separate manuscript. The current submission to Nature Communications is the only body of work under consideration for journal publication from this project. Unlike traditional methods however, deep learning systems have the capacity to improve as additional data is made available. We anticipate that with time, improved AUCs are a feasible target with our framework.

Reviewer #2 (Remarks to the Author):

The paper presents a deep learning system for predicting post-operative right ventricular failure (RV failure) using pre-operative echocardiography scans. The video-based deep model is trained end-to-end to directly predict the RV failure based on the spatiotemporal information of the echo videos.

The main strengths of the paper are:

- The manuscript is generally well written and properly organized.
- A very interesting clinical problem is tackled
- The system outperforms a team of human experts in the task of predicting RV failure based on echocardiograms.
- The presented details of the system in the manuscript is adequate and the methodology has high reproducibility.

However, the major limitations and possible spaces for improvement of the paper are:

1. The methodology is not fairly quite technically novel. The video-based deep models are extensively explored in the literature. In specific, two-stream video processing using optical flow as a secondary network input is previously explored in computer vision (e.g. Fig 2-e in [1]) and also in echo imaging [2]. Investigation of more recent trends in deep learning models for video analysis, such as [3, 4, 5] could add technical value to the paper. Analysis of recent deep learning Spatio-temporal models is given in [6].

[1] Carreira, Joao, and Andrew Zisserman. "Quo vadis, action recognition? a new model and the kinetics dataset." proceedings of the IEEE Conference on Computer Vision and Pattern Recognition. 2017.

[2] A Unified Framework Integrating Recurrent Fully-Convolutional Networks and Optical Flow for Segmentation of the Left Ventricle in Echocardiography Data, https://link.springer.com/chapter/10.1007/978-3-030-00889-5_4

[3] SlowFast Networks for Video

Recognition, https://openaccess.thecvf.com/content_ICCV_2019/papers/Feichtenhofer_SlowFast_Networks_for_Video_Recognition_ICCV_2019_paper.pdf

[4] Video Action Transformer

Network, https://openaccess.thecvf.com/content_CVPR_2019/papers/Girdhar_Video_Action_Transformer_Network_CVPR_2019_paper.pdf

[5] X3D: Expanding Architectures for Efficient Video

Recognition, https://openaccess.thecvf.com/content_CVPR_2020/papers/Feichtenhofer_X3D_Expanding_Architectures_for_Efficient_Video_Recognition_CVPR_2020_paper.pdf

[6] Deep Analysis of CNN-based Spatio-temporal Representations for Action

Recognition, <https://arxiv.org/pdf/2010.11757.pdf>

Thank you for this helpful review. We additionally explored a variant of the vision transformer networks described recently with kinetics600 pre-trained weights for a variety of tasks.⁷ On the Stanford AIMI EchoNet dataset of 10,000 echocardiograms, a 'divided space-time attention' vision transformer network trained from scratch achieved a mean absolute error of 6.7% whereas 3d residual networks achieved a MAE of ~ 5%. Video transformer networks that encapsulate some of the concepts of the 'Slow-Fast' neural networks have also recently been published (April 22nd on ArXiv).⁸ This remains a rapidly evolving area of research, with limited examples of adoption outside of Kinetics / ImageNet benchmarks. While we believe self-attention and transformer networks will have an important role to play in medical pre-training and downstream finetuning tasks, the transformer network variants we tested did not surpass convolutional networks for both the EchoNet task and the RV failure prediction task. This may be compounded by the fact that most vision transformers require intensive pre-training to learn features of importance, and self-supervised pre-training on medical datasets may be essential for such clinical problems.⁸⁻¹⁰ Our work with 3D convolutional neural network establishes a baseline for this clinical problem. We hope that advances in neural network architectures and training methods will improve on these baselines.

We have however included a list of neural network architectures that we tested (validation set results) available in

Supplementary Table 7, and shown below:

Model Architectures	AUC (95% CI)	Kinetics 600 pretrained
Resnet-18	0.533 (0.456 – 0.609)	No
Resnet-50	0.604 (0.319 – 0.890)	No
Resnet-152	0.556 (0.438 – 0.676)	No
Resnet-18	0.631 (0.554 – 0.709)	Yes
Resnet-50	0.702 (0.592 – 0.811)	Yes
Resnet-152	0.697 (0.589 – 0.804)	Yes
Resnet-18 + EHR	0.644 (0.525 – 0.762)	Yes
Resnet-50 + EHR	0.682 (0.617 – 0.746)	Yes
Resnet-152 + EHR	0.739 (0.642 – 0.837)	Yes
Resnet-18 + 224px input	0.673 (0.601 – 0.746)	Yes
Resnet-18 + 224px input	0.599 (0.522 – 0.675)	No
Resnet-50 + optical flow	0.686 (0.619 – 0.752)	Yes
Resnet-152 + optical flow	0.749 (0.634 – 0.863)	Yes
TimeSformer	0.547 (0.476 – 0.614)	Yes

2. The paper aims to predict RV failure solely based on pre-operative echo videos (apical 4-chambers view). In this setup, while the method outperforms clinical experts, it seems still the obtained accuracy may not be high; sensitivity is less than 50% at 80% specificity (reported AUC of 0.729, specificity of 52% at 80% sensitivity and 46% sensitivity at 80% specificity). It can be understood that the accuracy limitation could be due to the highly challenging nature of the tackled problem. The authors may further clarify the importance and clinical applicability of the obtained results.

We have expanded on the clinical importance and feasibility in the manuscript. Specifically, we elaborate on the role of early and aggressive therapy instituted based on AI tools such as ours that may help reduce the burden of post-operative RV failure and the healthcare expenses incurred. The relevant subsection titled “Echocardiography AI system and performance” reads:

“We evaluated the ML system at both a liberal operating point of 80% sensitivity, where the specificity was 52.75%, and at a more conservative operating point of 80% specificity, where the sensitivity was 46.67%. Balancing the risks and potential benefits of interventions in this patient population may require different thresholds of sensitivity and specificity. Observational data suggests that early and aggressive therapy with RVADs may improve survival in patients who are likely to develop post-operative RV failure.¹¹ These decisions must be made taking into account the significant additional morbidity associated with biventricular mechanical circulatory assist.^{12–14} Right ventricular assist devices potentially could thus be offered based on cutoffs from the conservative operating point of our ML system.”

We demonstrate the feasibility of tackling this challenging problem with AI, and expect that performance improvements may be possible with additional participating centers providing data, novel algorithms, and innovative pretraining strategies.

3. There is the possibility that more information may be needed to properly predict the RV failure. Possible improvements in the results may be obtained by investigating the incorporation of more information into model input, such as the addition of other echo views and other relevant info from the patient's health record.

This is a very relevant comment. Recent work in echocardiography AI has shown, it is possible that additional views may be needed for more accurate prediction¹⁵, however these views were not available in a consistent fashion in our retrospective dataset. At Stanford for example, less than 40% of the patients with apical 4 chamber views had a usable short axis view. Similarly, less than 20% of the patients with apical 4 chamber views had a usable

parasternal long axis view. More importantly, the different views were often available from scans taken at different days for the same patient – concatenating features from such disparate data without would be challenging given the moderate size of our dataset. This is however an important point, and something to be standardized for future prospective echo data collection protocols for LVAD patients. For researchers interested in this area, prospective collection of intra-operative transesophageal echocardiography might be an interesting alternative approach.

As for adding information from patient’s health record as an additional input, we performed the following additional experiments: To incorporate clinical biomarker and demographic information into the AI system, we incorporated the following: Gender {int}, admission INTERMACS score {int}, and pre-operative serum creatinine {float} and ALT {float}. The integer {int} variables were used as is, the numeric (float) variables were scaled and normalized across the entire dataset. We then concatenated a vector of dimensions {batch size, 4} containing this information to the terminal fully connected layer of 3 residual network architectures. The networks were trained as described elsewhere in the manuscript. We describe the results on the validation set below, and in supplementary table 7. The improvements were most obvious in the smaller residual networks with benefits that tapered off with deeper ones in the validation set.

Model with EHR data	AUC (95% CI)
Resnet-18	0.644 (0.525 – 0.762)
Resnet-50	0.682 (0.617 – 0.746)
Resnet-152	0.739 (0.642 – 0.837)

We have updated the Methods section under ‘Neural Network Architecture and Training’ to now read:

“The first convolutional layer utilizes a 7 x 7 x 7 kernel.¹⁶ The network weights were initialized using the Xavier normal initializing scheme, and was optimized using the AdamW algorithm.^{17,18} We incorporated clinical and demographic variables into the vision network by concatenating a 1-D vector to the terminal fully connected layer of each residual network. We showed marginal improvements over several pure vision-based neural networks in Supplementary Table 7. The network was trained for 50 epochs on a batch size of 8, with an initial learning rate of 1×10^{-5} .”

We have also amended the Discussion in Paragraph 4 to read as: *“As part of additional experiments, we evaluated the incorporation of certain baseline demographic and clinical features (Gender, INTERMACS score, Creatinine, ALT) into the training loop. We did not see a meaningful improvement over our best performing video-based system, though the improvements were more pronounced in some of the shallower neural networks evaluated (Supplementary Table 7). Additional hemodynamic information in the form of invasive pressure telemetry may further improve the performance of multi-modal AI systems integrating health record information and imaging.”*

Other minor comments:

4a. The input to the network is 32 frames of the video. The authors could clarify if the input videos are trimmed or re-sampled to 32 frames? If they are trimmed, what is the used criterion for selecting the input temporal window?

Thank you for this comment. We have clarified the methodology used for selecting the input temporal window. Specifically, our data generator takes an input video and returns an array for training of the size [frames, rows, columns, channels]. Where the starting frame is a randomly generated integer and from there onwards 32 sequential frames are taken. This process repeats itself meaning that at different epochs the specific temporal window may change. Anecdotally most scans were around 50 frames in length after temporal resampling to

normalize based on framerate. The relevant methods section under ‘Data pre-processing’ now reads:

“The frames of the processed videos were additionally down-sampled by bi-linear interpolation to a 112x112 resolution, and all videos were temporally normalized to framerate prior to training and evaluation. Additional experiments with an input resolution of 224x224 pixels were also conducted on shallower residual networks. Optical flow was calculated prior to model training using an OpenCV implementation of the Gunnar Farneback method based on polynomial expansion.⁵³ Additional data augmentation operations were performed on each frame of the videos in the following order as part of the training loop: random rotation up to 10°, random brightness multiplications, and random 2D shearing.”

4b. Adding a color legend to Fig. 3 could improve the clarity of the figure.

Thank you for this comment, we have updated Fig. 3 with a new visual confusion matrix and additional color legend (please see response to reviewer #3).

4c. The "LVAD" acronym is first used in line 45 and later defined on line 68.

We have added the definition for LVAD earlier in the manuscript prior to its first usage.

Reviewer #3 (Remarks to the Author):

The authors present a proper study of echocardiogram videos for an interesting prediction task: post-operative RV-failure from pre-operative data. The results while modest in terms of AUC, greatly outperform baselines presented. The methods seem to be an interesting contrast to Ouyang et al. Overall, I would recommend the paper for revision before acceptance.

Comments:

1. It would be nice to expand on Fig1b to explain to the audience how manual systems are currently employed in the clinical decision making workflow. As this is Nature Comms and not a Cardiology-specific journal, we shouldn't assume such knowledge. A natural question is how could the manual systems perform so poorly as standard-practice and what expectations should the reader have in terms of reasonable latent patterns in the data that one would expect could be useful for prediction?

We have updated Fig 1 now includes an additional diagram graphically describing the clinical workflow. We also added a new section in the paper titled “Clinical decision-making workflow for end stage heart failure” that has been described in an earlier response (above).

We hypothesized that manual metrics perform poorly since they are derived by measurements taken from one or two frames in the echocardiogram, where the rate of change of certain myocardial features may be important. We assessed the impact of reducing the number of frames that are processed by our neural networks. We found that reducing the number of frames on our best performing model led to a significant decline in performance to AUC 0.620 (0.546 – 0.696).

2. The authors mention "We tested a variety of approaches, including various pretraining strategies, optimizers, input streams, and model architectures." But there's no discussion on what was actually attempted and comparative results. In particular, the input streams + model architectures are of most interest.

Thank you for bringing attention to this important point. This is something other reviewers have also mentioned in their comments, and we have now included a list of neural network architectures and approaches that we tested in Supplementary Table 7 (also shown below). The table shows validation set performance for all of the following:

Model Architectures	AUC (95% CI)	Kinetics 600 pretrained
Resnet-18	0.533 (0.456 – 0.609)	No
Resnet-50	0.604 (0.319 – 0.890)	No
Resnet-152	0.556 (0.438 – 0.676)	No
Resnet-18	0.631 (0.554 – 0.709)	Yes
Resnet-50	0.702 (0.592 – 0.811)	Yes
Resnet-152	0.697 (0.589 – 0.804)	Yes
Resnet-18 + EHR	0.644 (0.525 – 0.762)	Yes
Resnet-50 + EHR	0.682 (0.617 – 0.746)	Yes
Resnet-152 + EHR	0.739 (0.642 – 0.837)	Yes
Resnet-18 + 224px input	0.673 (0.601 – 0.746)	Yes
Resnet-18 + 224px input	0.599 (0.522 – 0.675)	No
Resnet-50 + optical flow	0.686 (0.619 – 0.752)	Yes
Resnet-152 + optical flow	0.749 (0.634 – 0.863)	Yes
TimeFormer	0.547 (0.476 – 0.614)	Yes

3. The authors are directly feeding multiple frames as a 0-th dimension of the tensor to the ResNet. While Ouyang has a similar approach with highly curated inputs. The standard practice within video-processing in deep learning and computer vision has shown the importance of including some modeling inductive bias (e.g. recurrent neural network variants or positional encoding in transformers) to leverage the concept of time.

The spatial localization inductive bias in 3D convolutional neural networks extends to the temporal dimension, and since the echocardiography sequences are only 32 frames in length, we hypothesized that convolutional neural networks would suffice (as others have shown with this imaging modality). Beyond this clinical problem of interest, we expect true temporal inductive biases to be more relevant when concatenating results from scans taken at multiple follow-up visits. As for transformers, recent work with multiscale vision transformers has shown that current video vision transformers such as ViT-B actually function more akin to a bag-of-frames classifier rather than true spatio-temporal video classifiers.⁸ In a series of frame shuffling experiments, the authors show that for many vision transformers, performance remains stable even after shuffling the order of the frames in a video. Multiscale vision transformers (MVIT) on the other hand (an extension of the slow-fast paradigm to vision transformers) is sensitive to frame shuffling and therefore temporal positional embeddings are actually used for downstream classification. Transformers in general have few inductive biases, which might be why they require extensive (on large often proprietary datasets) pre-training to approximate some of the inductive biases of convolutional neural networks. Finally, incorporating specifically engineered temporal modules such as the ‘inflated temporal gaussian

mixture layer' from the 2019 EvaNet paper added only 1-2% to top-1 accuracy on the Kinetics400 dataset over traditional 3D convolutional neural networks.¹⁹

4. What type of data curation and data augmentation is or is not performed? this could greatly improve results and affect adoptability implications

As part of the training loop, we perform data augmentation operations such as scaling, rotation, brightness multiplication. We also implemented a 2D shearing procedure (same shearing transformation then repeated across each frame from a single video). We did not curate the dataset for scan quality or grain. We refrained from heavy curation to ensure the algorithms remained adoptable in the real world.

5. Captions need to be revised. They're currently very broadly written and not explanatory enough. authors should include key active takeaways and point out / explain the most relevant comparisons/acronyms/names

We have amended all the captions to include additional information for each Figure to now read as follows:

Fig. 1. Overview of the project: *a* Pre-operative echocardiography videos are processed as a stack of 32 frames. A two-stream implementation of raw greyscale videos and optical flow channels are fed into a 3D convolutional neural network to produce the prediction of RV failure *b*. The clinical workflow for predicting future risk of RV failure begins in the pre-operative phase using a combination of clinical parameters and a detailed echocardiographic assessment. Risk scores such as the CRITT and Penn scores are calculated thereafter to aid in risk stratification following which a decision is made to either electively implant a concomitant RVAD or proceed with LVAD alone. *c*. The clinical ground truth is determined largely by the persistent need for inotropes past post-operative day 14 or right ventricular mechanical circulatory assist devices during the post-operative recovery period. MCS-ARC definitions are detailed in Supplementary Table 1.

Fig. 2 Performance of the AI system, clinical risk scores, and clinical benchmarking. *a*. ROC curve of the AI system compared to contemporary clinical risk scores. The performance of the AI system was 0.729 (95% CI 0.623-0.835). *b*. ROC curves of clinical expert team and independently calculated metrics of right ventricular function compared to the AI system. The performance of the AI system was found to exceed both clinical experts and the traditional risk scoring systems. Abbreviations: LV-ESA (Left ventricular end systolic area), RVED-Area (Right ventricular end diastolic area), RVEF (RV Ejection Fraction), RVES-Area (RV end systolic area).

Fig. 3: Analysis of saliency maps from pre-operative echocardiograms. Representative input videos and visualizations for both systolic and diastolic phases of the cardiac cycle across patients with and without RV failure, in the form of a confusion matrix. True positives (bottom right quadrant), false positives (bottom left quadrant), true negatives (top left quadrant), and false negative examples (top right quadrant). Color scale for each quadrant represents regions that contributed most to the predicted class (red) and those that pushed predicted probability away from the predicted class (blue).

Additionally, certain legends in the supplementary appendix have also been revised.

6. There needs to be more analysis broadly speaking. In particular Figure 3 could benefit from additional analysis.. for example there could be a confusion matrix to peer into when/why the model and manual systems are right vs wrong (with some interpretation as well)

Thank you for this comment, we have amended Figure 3 to be a confusion matrix visualization. The threshold for prediction was kept at 0.5 when creating the confusion matrix. The saliency map section is now expanded to include interpretations for correct vs incorrect predictions. The last few lines of the 'Saliency maps and Visualizations' section now reads:

"For patients where the AI system correctly predicted RV failure, saliency maps localized over the right atrium and the region of the tricuspid annulus. Similarly, the AI system correctly predicted the absence of RV failure when activation was localized over the right ventricle and right atrium. In cases where the AI system appeared to rely heavily on the interatrial and interventricular septum, the quality of predictions appeared to decline. While septal motion aberrations are often been described as a predictor of RV failure in the clinical setting, they may also be a feature of acute isolated RV volume overload – a challenging overlap in presentation on echocardiography.⁴²"

Minor comments:

7a. Figure 1 caption needs to be revised to have a one-liner followed by A and B.

Thank you for these comments, we have amended the caption for Figure 1 to now read:

"Fig. 4. Overview of the project:"

7b. Improve background to cite related work in deep learning and echo:

- Madani, Ali, et al. "Fast and accurate view classification of echocardiograms using deep learning." NPJ digital

medicine 1.1 (2018): 1-8.

- Ghorbani, Amirata, et al. "Deep learning interpretation of echocardiograms." *NPJ digital medicine* 3.1 (2020): 1-10.

We have added references for the work by Madani *et al.* and Ghorbani *et. al.* in the discussion section. The relevant sections are highlighted below in Paragraph 3 of the discussion section.

"All contemporary echocardiography ML systems rely on supervised segmentation algorithms to outline cardiac chambers. Most recently, Ouyang et al described a weakly supervised video segmentation system to calculate ejection fraction using left ventricular tracings in conjunction with spatiotemporal convolutions.^{30,47"}

And later in Paragraph 4:

"Madani et. al describe an image-based neural network for automated echocardiography view classification that could be integrated into a multi-view RV failure risk prediction system, and it is plausible that superior performance can be attained using prospectively collected echocardiographic data from multiple view planes.^{52"}

7c. Wonder if it makes more sense to pretrain on EchoNet dynamic dataset instead to enable better transfer learning. for discussion on usage of natural images for medical imaging prediction tasks, refer to Ke, Alexander, et al. "CheXtransfer: performance and parameter efficiency of ImageNet models for chest X-Ray interpretation." arXiv preprint arXiv:2101.06871 (2021).

We attempted to transfer learn from EchoNet dynamic and found the results were nearly identical to networks initialized with random weights. This was despite converging to a solution that were close to the published results for this regression problem (mean absolute error of 5%). We conjecture that this may be due to some form of shortcut learning on the EchoNet dataset regression problem, that fails to generalize well for RV failure prediction, though we did not explore why this could be the case. There may be opportunities to leverage the EchoNet dataset via self-supervised pretraining techniques (student-teacher / contrastive learning / DINO), but such a comparison would be beyond the scope of this current manuscript.^{20,21} Our results represent those of contemporary methods with supervised learning, and may serve as a baseline for further improvements via self-supervised pre-training with datasets within the same imaging / anatomical domain. We have added a small section in the discussion elaborating on such future directions. The relevant lines in paragraph 4 now read:

"We found that pre-training with large video datasets (Kinetics-600) was critical for model performance.³² The same could not be said for pre-training from the Echo-Net Dynamic dataset despite sharing the same imaging modality. While beyond the scope of this current manuscript, future attempts at self-supervised pre-training with cardiovascular imaging datasets may yield superior results.^{54,54} Finally, unlike hard radiological or histopathological ground truth labels, the clinical ground truth for post-operative RV failure leaves room for subjectivity despite the MCS-ARC guidelines."

Reviewer #4 (Remarks to the Author):

1. I have experience using optical flow/two stream networks in echocardiography also, and actually my personal experience was the optical flow aren't didn't add _that_ much beyond either 3D CNNs or recurrent convolutional CNNs. Did the authors investigate other forms? My suspicion is that the optical flow algorithms start to break down a bit due to significant echocardiographic noise and the general textured nature of the myocardium.

Thank you for your comment. Anecdotally we have not found optical flow to be useful in other tasks (EchoNet Dynamic for example). We have added additional experiments that show the relative improvement with optical flow vs without for this specific problem. We suspect that the benefit of optical flow exists because of the sharp demarcation between the cardiac chamber and myocardial wall (in terms of pixel intensity) and our relatively smaller dataset size. We saw improvements (albeit inconsistently) across shallow and deep residual networks

when using additional optical flow data. Incorporating clinical variables also helped but combining both EMR and Flow data did little to improve performance beyond what either approach achieved alone.

2. In the data augmentation, the authors say they used "3-dimensional shearing". Do they mean they used 2d shearing on each frame, or they literally sheared in 3 dimensions? The latter sounds incredibly disruptive, as you'd be moving spatial data from 1 frame into another, essentially introducing wild levels of dyskinesia, with systole in 1 part of the heart moving into a diastolic period of another, for example. "Normal" hearts are unlikely to look normal after that, surely, and picking up the subtle signs of septal motion which they postulate are important but be next to impossible?

Thank you for this important comment, we have added clarification to this: We performed a random 2D shearing procedure on the first frame that was then repeated on each frame of a video. We have amended the methods section in the 'Data pre-processing' paragraph to include:

"Additional data augmentation operations were performed on each frame of the videos in the following order as part of the training loop: random rotation up to 10°, random brightness multiplications, and random 2D shearing"

3. I'm surprised they had to use videos as small as 112 * 112 pixels, as the original Quo Vadis paper many years ago I think used similar sized videos, and obviously GPU sizes have progressed significantly since then. I suppose this is because the authors used a _VERY_ deep (152 layers) ResNet rather than the VGG networks/13D used previously. I personally wonder if using a less deep network but with higher resolution might be better, and would be interested to hear if the authors tried this.

We tried using 224 x 224 videos in addition to the 112 x 112-pixel size for a much shallower Kinetics600 pretrained 18-layer neural network. We found that performance does increase with the larger frame sizes (AUC of 0.673 (0.601 – 0.746) vs 0.631 (0.554 – 0.709)), though performance remained worse than the deeper neural networks. We detail this along with the other experiments attempted in Supplementary Table 7.

We have amended the methods section to include additional details about the pretraining and input resolution.

4. My personal feeling is standard formulae such as BCEloss do not need to be given in a manuscript where 1) anyone who is interested in that level of depth would already know it, 2) it's not the focus of the paper. But I don't feel too strongly.

We have amended the methods section of the paper and have removed the equation for BCE loss.

5. My personal feeling is pretraining on non-medical datasets before transferring to completely different modalities is overhyped and in my experience helps little, versus just good initialisation, See <https://arxiv.org/abs/1902.07208>. But I accept this is a somewhat controversial view and appreciate most reviewers will expect it. I am therefore interested to hear the authors found it useful.

We have added additional details to the supplementary appendix detailing the pre-training / transfer learning for different neural network architectures. We appreciate this comment and the link to the interesting article by Raghu *et al.*

Pre-training on ImageNet / Kinetics may very well be a relatively straightforward way to optimize initialization, and there indeed is conflicting research around this – most recently in a paper on contrastive pre-training, Dr. Langlotz's (co-author) had these findings for example²¹: *"We also notice that our conclusion of using ImageNet versus random initialization is different from Raghu et al. (2019): while they showed comparable results from the two strategies, we find that using ImageNet initialization is still superior than random initialization in most results, justifying its popularity. Upon closer examination, we conjecture that this is likely due to under-optimization of their models: while our ResNet50 with random initialization achieves an average AUC of 85.8 on the CheXpert dataset, their ResNet50 model only achieved 83.5 AUC on the same evaluation set."*

References:

1. Lampert, B. C. & Teuteberg, J. J. Right ventricular failure after left ventricular assist devices. *Journal of Heart and Lung Transplantation* **34**, 1123–1130 (2015).
2. Murthy, S. C. & Blackstone, E. H. Commentary: We prefer wisdom over knowledge. *The Journal of Thoracic and Cardiovascular Surgery* **161**, 1942–1943 (2021).
3. Tomašev, N. *et al.* A clinically applicable approach to continuous prediction of future acute kidney injury. *Nature* **572**, 116–119 (2019).
4. Zaidi, A. *et al.* Echocardiographic assessment of the right heart in adults: a practical guideline from the British Society of Echocardiography. *Echo Research and Practice* **7**, G19–G41 (2020).
5. Rudski, L. G. *et al.* Guidelines for the echocardiographic assessment of the right heart in adults: a report from the American Society of Echocardiography endorsed by the European Association of Echocardiography, a registered branch of the European Society of Cardiology, and the Canadian Society of Echocardiography. *J Am Soc Echocardiogr* **23**, 685–713; quiz 786–788 (2010).
6. Ammar, K. A. *et al.* The ABCs of left ventricular assist device echocardiography: a systematic approach. *European Heart Journal - Cardiovascular Imaging* **13**, 885–899 (2012).
7. Bertasius, G., Wang, H. & Torresani, L. Is Space-Time Attention All You Need for Video Understanding? *arXiv:2102.05095 [cs]* (2021).
8. Fan, H., Xiong, B. & Mangalam, K. Multiscale Vision Transformers. *arXiv:2104.11227 [cs.CV]* 18 (2021).
9. Neimark, D., Bar, O., Zohar, M. & Asselmann, D. Video Transformer Network. *arXiv:2102.00719 [cs]* (2021).
10. Dosovitskiy, A. *et al.* An Image is Worth 16x16 Words: Transformers for Image Recognition at Scale. *arXiv:2010.11929 [cs]* (2020).
11. Raymond Fitzpatrick, J. *et al.* Early planned institution of biventricular mechanical circulatory support results in improved outcomes compared with delayed conversion of a left ventricular assist device to a biventricular assist device. *The Journal of Thoracic and Cardiovascular Surgery* **137**, 971–978 (2009).
12. Kiernan, M. S. *et al.* Early Right Ventricular Assist Device Utilization in Patients Undergoing Continuous-Flow Left Ventricular Assist Device Implantation: Incidence and Risk Factors from INTERMACS. *Circulation. Heart failure* **10**, (2017).
13. Kormos, R. L. *et al.* The Society of Thoracic Surgeons InterMACS Database Annual Report: Evolving Indications, Outcomes, and Scientific Partnerships. *The Annals of thoracic surgery* **107**, 341–353 (2019).
14. Morgan, J. A., John, R., Lee, B. J., Oz, M. C. & Naka, Y. Is severe right ventricular failure in left ventricular assist device recipients a risk factor for unsuccessful bridging to transplant and post-transplant mortality. *Annals of Thoracic Surgery* **77**, 859–863 (2004).
15. Ulloa Cerna, A. E. *et al.* Deep-learning-assisted analysis of echocardiographic videos improves predictions of all-cause mortality. *Nature Biomedical Engineering* 1–9 (2021) doi:10.1038/s41551-020-00667-9.
16. Hara, K., Kataoka, H. & Satoh, Y. Learning Spatio-Temporal Features with 3D Residual Networks for Action Recognition. *arXiv:1708.07632 [cs]* (2017).
17. Glorot, X. & Bengio, Y. Understanding the difficulty of training deep feedforward neural networks. in *Proceedings of the thirteenth international conference on artificial intelligence and statistics* 249–256 (2010).
18. Kingma, D. P. & Ba, J. L. Adam: A method for stochastic optimization. in *3rd International Conference on Learning Representations, ICLR 2015 - Conference Track Proceedings* (International Conference on Learning Representations, ICLR, 2015).
19. Piergiovanni, A., Angelova, A., Toshev, A. & Ryoo, M. Evolving Space-Time Neural Architectures for Videos. in

2019 IEEE/CVF International Conference on Computer Vision (ICCV) 1793–1802 (IEEE, 2019).
doi:10.1109/ICCV.2019.00188.

20. Caron, M. *et al.* Emerging Properties in Self-Supervised Vision Transformers. *arXiv:2104.14294 [cs]* (2021).
21. Zhang, Y., Jiang, H., Miura, Y., Manning, C. D. & Langlotz, C. P. Contrastive Learning of Medical Visual Representations from Paired Images and Text. *arXiv:2010.00747 [cs]* (2020).

Reviewers' Comments:

Reviewer #1:

Remarks to the Author:

I am satisfied with the authors' response. no further comments from my side.

Reviewer #2:

Remarks to the Author:

Thank you for the responses. The authors have adequately addressed my concerns.

Reviewer #3:

Remarks to the Author:

The authors have addressed most, if not all, of my prior comments. I think the additional subfigures, supplementary tables, and revised text/captions strongly improves the manuscript.

Reviewer #4:

Remarks to the Author:

The authors have responded to the reviewers' questions very well and dealt with my concerns.